# Sustainable Investment Decision-Making on Office Buildings using Reinforcement Learning and Large Language Models

**ChatGPT-5**[1]    **Ziru Tao**[2]*    **Paul Baguley**[2]    **Rashid Maqbool**[2]    **Obuks Ejohwomu**[2]

[1]OpenAI, 1455 3rd Street, San Francisco, CA 94158, United States.
[2]Department of Civil Engineering and Management, School of Engineering, The University of Manchester, Manchester, M13 9PL, United Kingdom.

## Abstract

This study develops a reinforcement learning (RL) framework to optimize lifecycle investment decisions for sustainable office buildings from a cost engineering perspective, translating Environmental, Social, and Governance (ESG) impacts into monetized drivers for decision support. Sequential choices across design, construction, and operation are modeled as a Markov Decision Process (MDP) and trained with a Deep Q-Network, aligning the discount factor with the economic discount rate to avoid double counting. A large language model (LLM), ChatGPT-5, is used to extract parameters from unstructured guidance and to generate stakeholder-facing explanations of learned policies. Across two case studies in the United States and the United Kingdom, the RL strategy achieves 37.5–45.0% lower annual energy use and 31.0–36.9% lower total lifecycle carbon than conventional practice. Despite a 4–6% higher initial cost, it reduces financial lifecycle cost by $0.42 million (US) and £1.01 million (UK) and reduces societal cost NPV, i.e., monetized carbon and productivity effects, by $3.50 million (US) and £3.00 million (UK). Results remain robust under $\pm 20\%$ parameter noise and a $+2°C$ climate scenario. Limitations include reliance on secondary estimates for social valuation, simplified transition dynamics, and automated evaluation of LLM explanations; future work will incorporate expert blind review and real project validation.

**Keywords:** Reinforcement Learning; Cost engineering; Environmental, Social, and Governance (ESG); Markov decision process (MDP); Office buildings; Large language model (LLM).

## 1 Introduction

The building sector accounts for approximately one third of global greenhouse gas emissions and 40% of final energy consumption in regions such as the European Union [1]. In the United States alone, commercial buildings consumed 6.8 quadrillion Btu of energy in 2018, emphasizing both environmental impacts and significant financial stakes [2]. Beyond energy and carbon, sustainable building investment increasingly incorporates broader environmental, social, and governance (ESG) criteria, including occupant health, productivity, and community benefits [3]. However, conventional investment practices often prioritize short-term financial returns, undervaluing long-term sustainability benefits due to difficulties in quantification and integration into decision-making processes [4, 5].

Cost engineering is central to aligning sustainability with financial feasibility, yet facilities management (FM) practice faces six structural challenges that currently limit trustworthy Artificial Intelligence (AI)-driven ESG decision-making. First, there is a scarcity of documented, end-to-end cases that integrate AI and ESG within FM portfolios, making it hard to establish credible precedents [1]. Second, standardization is weak across data schemas, model interfaces, and reporting conventions

---

*Corresponding author: ziru.tao@manchester.ac.uk

at the intersection of AI and cost engineering, which undermines comparability and audit trails [6, 7]. Third, guidance is limited on how the cost engineer's role evolves from estimator to steward of ESG-monetized value and model governance [7, 8]. Fourth, public benchmarks, e.g., prices, grid carbon intensity, are rarely linked systematically to private project or portfolio data, impeding external validation and transfer learning [9–12]. Fifth, large language models (LLMs) remain loosely coupled to cost-estimating workflows (parameter extraction, assumption tracing, narrative justifications), so their value is under-realized [13, 14]. Sixth, there is no systematic, transparent procedure for selecting and justifying AI model structural elements, especially reward architectures and preference weights, so results are not easily auditable or reproducible [15–17].

Life-cycle cost analysis (LCCA) and multi-criteria decision analysis (MCDA) are well-established methods for integrating sustainability into investment decisions. LCCA accounts for long-term financial impacts [18], while MCDA balances economic, environmental, and social criteria. When combined, these tools offer robust mechanisms for evaluating complex trade-offs across a building's life-cycle [19]. Applications of LCCA-MCDA frameworks have successfully incorporated resilience metrics, social values, and environmental impact categories such as global warming potential and primary energy use [20].

Additionally, Maddaloni and Sabini [4], Burchart and Przytuła [20], Du et al. [21] demonstrate that stakeholder-informed weighting, uncertainty modeling, and visualized scoring can enhance transparency and decision quality. However, these methods remain underutilized in sequential, dynamic decision-making contexts where sustainability values evolve over time.

Recent progress offers ingredients to address these issues. Reinforcement learning (RL) has shown strong performance in building operations, but most applications focus narrowly on short-horizon control rather than life-cycle, multi-objective investment planning that reflects ESG value [1, 16]. In parallel, LLMs can operationalize unstructured guidance, surface provenance, and generate stakeholder-facing rationales; however, they are seldom wired directly into cost-estimating pipelines to extract parameters, reconcile public and private sources, or produce auditable explanations of AI-driven decisions [13, 14].

Consequently, the key gaps motivating this study are: (i) few end-to-end FM cases that join AI with ESG outcomes [1]; (ii) limited standardization between AI methods and cost-engineering practice [6, 7]; (iii) insufficient guidance on the evolving role of the cost engineer in AI/ESG model governance [8]; (iv) weak linkage of public benchmarks with private datasets for validation and transfer [9–12]; (v) minimal integration of LLMs with AI-based cost estimating [13, 14]; and (vi) no systematic method to choose and justify model structure—especially the reward function and weights—in life-cycle RL for buildings [15–17, 20, 22].

To address these gaps, this study aims to develop and validate a dynamic decision-making framework for sustainable office building investment that integrates RL, LLMs, and cost engineering. Three objectives are: (1) quantifying social values using established monetization, e.g., Social Cost of Carbon (SCC) on real-world cases; (2) comparing an RL-based strategy with conventional models in the US and UK; and (3) using LLMs to extract parameters from unstructured documents and produce stakeholder-facing explanations. Theoretically, the paper establishes two guarantees relevant to cost engineering governance: (i) an NPV–RL equivalence showing that aligning the per-stage discount factor with the economic discount rate yields the same optimal policy as minimizing life-cycle NPV with ESG adders; and (ii) a Pareto-support result showing that linear scalarization in the adopted Multi-Objective Reinforcement Learning (MORL) setup returns supported Pareto-efficient solutions for cost–carbon trade-offs. Practically, it also offers guidance on linking public and private data and on transparent reward/weight selection for auditability and reproducibility.

## 2 Methods

### 2.1 Case study protocol

This study investigates sustainable investment decisions for mid-sized office buildings around 10,000 square meters across three project phases: design, construction, and operation. The unit of analysis is the full life cycle, assessed over a 20-year operational horizon. Two empirical case studies, one in Chicago, USA, and the other in Manchester, UK, are selected due to their rich data availability and contrasting policy contexts. Both follow a standard project timeline: design in year 0, construction in

year 1, and operation from years 2 to 21 within 20 annual steps. A 3% discount rate is applied to reflect public sector investment norms.

The case study protocol aligns with established methodological guidelines. Yin [23] emphasizes that case studies are suitable for "how" and "why" research questions involving complex, real-life phenomena. This study adopts a multiple-case embedded design to explore cross-context performance of the RL-based strategy. It employs a deductive approach to test a theoretically informed model across contrasting contexts, enhancing external validity through replication logic [24].

This protocol ensures analytical generalization by demonstrating how the RL framework performs across diverse settings, while the structured comparison enables evaluation of both financial and societal outcomes, consistent with best practices in design science and policy-oriented case research [23, 24].

## 2.2 Model rationale

The research employs a reinforcement learning (RL) framework, formally modeled as a Markov Decision Process (MDP)[15]. It captures sequential, state-dependent decisions with stochastic transitions, ideal for life-cycle modeling, while other static optimization methods like LCCA or linear programming cannot model interdependent stages or uncertainty over time [15]. The MDP is defined by the tuple $(S, A, T, R, \gamma)$, where $S$ represents the state space, $A$ denotes the action space, $T$ describes the state transition dynamics, $R$ is the reward function, and $\gamma$ is the discount factor. We set $\gamma = e^{-r\Delta t}$ with $r = 0.03$ and $\Delta t = 1$ year for all decision steps in the 20-year operation horizon, yielding $\gamma \approx e^{-0.03} \approx 0.97045$.

The state space $S$ is designed to capture all relevant information at each project stage. A state $s \in S$ is a composite vector in Equation(1):

$$s = (s_p, s_d, s_c, s_b, s_x) \tag{1}$$

where $s_p$ is the categorical project stage (0: Pre-design, 1: Design, 2: Construction, 3: Operation), $s_d$ is a vector of chosen design features, e.g., insulation level, HVAC efficiency, pursuit of certifications, $s_c$ represents construction attributes, e.g., material selections, waste management practices, $s_b$ comprises current building performance metrics, e.g., predicted Energy Use Intensity (EUI), water use, carbon emissions, $s_x$ captures external context, e.g., climate zone, grid carbon intensity, utility rates, and local unemployment rate.

The action space $A$ consists of discrete investment choices available at each stage. (1) The design stage includes conventional design for code minimum, green design for improved efficiency plus 2% cost), ultra-green design for net-zero energy ready plus 5% cost. (2) The construction stage includes standard practice, green construction with low-carbon materials, enhanced social practice, i.e., local hiring, safety investments. (3) The operation stage includes standard facility management, smart energy management, wellness program.

Transition dynamics $T$ are modeled using a combination of building performance simulation data, empirical studies, and engineering assumptions. For instance, selecting a Green design action reduces predicted EUI by approximately 25% compared to a conventional baseline [16]. Construction actions primarily affect embodied carbon and social metrics, e.g., local job creation, while operational actions determine realized energy performance and occupant outcomes. Stochasticity is incorporated by adding $\pm 10\%$ noise to energy outcomes to account for uncertainties in weather and occupant behavior.

The reward function $R$ is architected to encapsulate the study's triple-bottom-line objective, integrating financial, environmental, and social value into a single scalar. The reward at the operation state is calculated in Equation(2):

$$r_t = -c_t + \sum_{i=1}^{n} \alpha_i f_i(s_t) \tag{2}$$

where $r_t$ is the immediate reward at discrete and yearly variable, time $t$, and $c_t$ is the undiscounted cash outflow incurred at $t$ including design, construction, and operational expenditure. The discounted return is $\sum_{t=0}^{T-1} \gamma^t r_t$ with $\gamma = e^{-r\Delta t}$. Net present value (NPV) is recovered externally as

NPV $= \sum_t e^{-rt\Delta t} c_t$, so discounting is applied once. Intermediate rewards equal the negative of the immediate cost plus any instantaneous social value, e.g., job-years realized at that $t$.

Additionally, a weight-conditioned Multi-Objective Reinforcement Learning (MORL) setup is adopted to address cost–carbon trade-offs presented in Figure 1. Its training framework and hyperparameters are detailed in appendices.

**Deep Q-Network**  A Deep Q-Network (DQN) algorithm is implemented to train the RL agent. It efficiently approximates Q-values in large, discrete action spaces using deep neural networks [25]. By contrast, policy gradient or actor-critic methods are less stable for discrete actions and require more hyperparameter tuning [15]. The neural network approximating the Q-function has an input layer matching the state dimension with about 20 features after one-hot encoding, two hidden layers with 64 neurons each using ReLU activation, and an output layer with nodes for each possible action [25]. The learning objective is to minimize the loss function $L(\theta)$ for the network parameters $\theta$ in Equation (3):

$$L(\theta) = \mathbb{E}_{(s,a,r,s')\sim U(D)} \left[ \left( r + \gamma \max_{a'} Q(s',a';\theta^-) - Q(s,a;\theta) \right)^2 \right] \tag{3}$$

where $\theta^-$ are the parameters of a target network, updated periodically, and $U(D)$ is a uniform distribution over the experience replay buffer $D$. Training employs an epsilon-greedy policy ($\epsilon$ decayed from 1 to 0.1), a replay buffer of 10,000 experiences, a batch size of 64, and the Adam optimizer with a learning rate of 0.001. It combines the advantages of Adaptive Gradient Algorithm (AdaGrad) and Root Mean Square Propagation (RMSProp) to achieve fast, adaptive convergence in noisy environments [26]. However, Stochastic Gradient Descent (SGD) and its variants require careful manual tuning and perform poorly with sparse gradients or non-stationary objectives [25]. To avoid double counting, we align $\gamma$ with the economic rate via $\gamma = e^{-r\Delta t}$; with $r = 0.03$ and $\Delta t = 1$ year this gives $\gamma \approx 0.97045$. Figure 2 reports sensitivity to reasonable $r$.

**Assumptions and theoretical results**  We state mild assumptions used throughout: (A1) finite horizon $T$ with bounded per-stage financial and ESG flows; (A2) within-stage stationarity of the transition kernel; (A3) a constant economic discount rate $r$ per stage $\Delta t$; (A4) reward scalarization uses fixed nonnegative weights; (A5) per-period cash flows $c_t$ are modeled undiscounted; discounting is applied only via $\gamma^t = e^{-rt\Delta t}$ in the return.

**Proposition 1** (NPV–RL equivalence). *Under (A1)–(A5), with $\gamma = e^{-r\Delta t}$ and the scalar reward in Eq. (2), maximizing the expected discounted return coincides with minimizing life-cycle NPV with ESG adders; the sets of optimal policies are identical.*

Unroll the Bellman recursion and collect per-stage flows: $\sum_{t=0}^{T-1} \gamma^t(-\text{Cost}_t + \sum_i \alpha_i f_i(s_t))$ is exactly the discounted-cash-flow objective with economic rate $r$ since $\gamma^t = e^{-rt\Delta t}$. Thus the dynamic-programming argmax equals the NPV argmin with ESG adders [15].

**Proposition 2** (Supported Pareto optimality under linear scalarization). *Under (A1)–(A4), any policy that maximizes a linear scalarization of the vector return (cost and carbon) for some nonnegative weight $w$ is Pareto-efficient (supported). Conversely, any supported Pareto point is optimal for some $w$.*

Optimality of $w^\top \mathbf{G}(\pi)$ implies no feasible policy strictly improves all objectives; supported points lie on the upper convex envelope and admit a supporting hyperplane defined by $w$. The dynamic nature is immaterial to the dominance argument because $\mathbf{G}(\pi)$ aggregates per-stage rewards and (A1)–(A2) ensure feasibility within the policy class [15].Full proof is presented in appendices and a flow diagram demonstrates an overview of the end-to-end model development pipeline in Figure 3.

**Temporal resolution and discounting**  Design (year 0) and construction (year 1) are modeled as single annual steps. Operation is modeled as 20 annual steps (years 2–21). Thus $T = 22$ steps in total. All cash flows $c_t$ are allocated at the end of each year $t$ and remain undiscounted inside $r_t$; discounting enters only through $\gamma^t = e^{-0.03t}$.

## 2.3  Data resource

To ensure transparency and reproducibility, this study uses publicly accessible datasets and peer-reviewed sources across five core categories: building energy use, construction and operational costs,

carbon emissions, occupant health and productivity, and social impact metrics such as job creation and safety. Data sources include the United States Department of Energy's Commercial Buildings Energy Consumption Survey, the Chartered Institution of Building Services Engineers benchmarks in the United Kingdom, RSMeans construction cost database, the UK Department for Business, Energy and Industrial Strategy, the World Green Building Council, and workplace safety statistics from the United States Occupational Safety and Health Administration and the UK Health and Safety Executive. These standardized and authoritative sources enable replication and adaptation across different regions and contexts. Full parameters and assumptions are listed in Table 1.

**Table 1:** Key Case Study Parameters and Data Sources

| Parameter | US Case | UK Case | Data Source |
|---|---|---|---|
| Climate & Design Days | 5400 HDD, 1100 CDD | 2400 HDD, 80 CDD | [27], [12] |
| Electricity Price | $0.10 per kWh | £0.18 per kWh | [9], [10] |
| Grid Carbon Intensity | 0.42 kg $CO_2$e/kWh | 0.25 kg $CO_2$e/kWh | [11], [12] |
| Baseline Energy Use | 240 kWh/m²/yr | 180 kWh/m²/yr | [9],[10] |
| High-Perf. Design | 150 kWh/m²/yr | 120 kWh/m²/yr | [28], [3] |
| Embodied Carbon (baseline) | 500 kg/m² | 400 kg/m² | [11], [29] |
| Embodied Carbon Reduction (green) | 15% | 20% | [9], [6] |
| Construction Cost | $2,200/m² | £1,800/m² | [30], [31] |
| Design Premium | +2% / +5% of constr. cost | +3% / +6% of constr. cost | [3] |
| Social Cost of Carbon | $190/ton | £160/ton | [11], [28] |
| Productivity Gain | 5% increase | 4% increase | [3] |
| Value of 1% productivity | $400 per employee-year | £250 per employee-year | [32] |
| Job Creation (baseline) | 10 jobs per $1M | 12 jobs per £1M | [33], [34] |
| Job Creation (enhanced) | 15 jobs per $1M | 14 jobs per £1M | [33], [34] |
| Accident Rate (baseline) | 3 per 200k hours | 1 per 100k hours | [35], [36] |
| Accident Rate (safety) | <1 per 200k hours | 0.5 per 100k hours | [35], [36] |

Units: energy intensity in kWh/m²/year; grid intensity in kg $CO_2$e/kWh. HDD/CDD denote heating/cooling degree days. Monetary values in 2023 prices unless noted.

## 2.4 Model training

The model environment is implemented as a stochastic simulation of a full building life-cycle, from design to operation. All numerical state variables are normalized, and categorical features are one-hot encoded to ensure training stability. The reinforcement learning agent is trained using a Deep Q-Network (DQN), with episodes simulating decision sequences over the building's life cycle.

Two baselines are used: (i) a financial-only agent optimizing $R = -\text{NPV}_{\text{cost}}$ and (ii) a heuristic "lowest first-cost" rule. Small-scale MCDA and MILP heuristics are additionally reported to show consistency on simple subproblems. Ablations vary network width (32–64–128), target-update period, replay size, and reward weights $\alpha_i$ to examine stability and the influence of social-value monetization.

The integration of LLM, ChatGPT 5, occurs in two distinct, non-training loops. On the one hand, unstructured textual resources, e.g., green building guidelines and ESG metric descriptions, are fed to the LLM with engineered prompts, e.g., "According to UKGBC, what is the typical energy savings of green offices?", to extract and validate numeric parameters for the model. On the other hand, after the RL agent generates a strategy for a case study, the sequence of decisions and outcomes is formatted into a prompt template asking the LLM to provide a step-by-step, natural language rationale for the AI's choices, mimicking an expert report to stakeholders.

**Table 2:** Comparative results for US case between RL and Conventional

| Metric | Conventional Strategy | RL Strategy | Difference |
|---|---|---|---|
| Initial Cost ($ million) | 22.00 ± 0.50 | 22.88 ± 0.55 | 0.88 |
| Annual Energy Use (kWh/m²) | 240 ± 12 | 150 ± 8 | –37.5% |
| Annual Energy Cost ($) | 240,000 ± 12,000 | 150,000 ± 7,500 | –90,000 |
| 20-yr Energy Cost NPV ($ million) | 3.47 ± 0.17 | 2.17 ± 0.11 | –1.30 |
| Annual Operational carbon (tons) | 1,000 ± 50 | 600 ± 30 | –40.0% |
| Embodied carbon (tons) | 5,000 ± 250 | 4,250 ± 200 | –15.0% |
| Total 20-yr carbon (tons) | 25,000 ± 1,250 | 17,250 ± 850 | –31.0% |
| Productivity Improvement (%) | 0 ± 0 | 3.5 ± 0.2 | 0.035 |
| Productivity NPV ($ million) | 0 | 10.11 ± 0.05 | 1 |
| Job-Years Supported | 500 ± 25 | 510 ± 26 | 10 |
| Life-cycle Cost NPV ($ million) | 25.47 ± 1.10 | 25.05 ± 1.05 | –0.42 |
| Societal cost NPV ($ million) | 22.00 ± 1.10 | 18.50 ± 0.93 | –3.50 |

Difference is RL minus Conventional. For cost metrics, a negative value indicates a reduction. Values are mean ± st.dev. across random seeds ($n \geq 5$).

**Table 3:** Comparative results for UK case between RL and Conventional

| Metric | Conventional Strategy | RL Strategy | Difference |
|---|---|---|---|
| Initial Cost (£ million) | 18.00 ± 0.90 | 19.10 ± 0.96 | 1.10 |
| Annual Energy Use (kWh/m²) | 180 ± 9 | 99 ± 5 | –45.0% |
| Annual Energy Cost (£) | 324,000 ± 16,200 | 178,200 ± 8,910 | –145,800 |
| 20-yr Energy Cost NPV (£ million) | 4.68 ± 0.23 | 2.57 ± 0.13 | –2.11 |
| Annual Operational carbon (tons) | 450 ± 23 | 250 ± 13 | –44.4% |
| Embodied carbon (tons) | 4,000 ± 200 | 3,200 ± 160 | –20.0% |
| Total 20-yr carbon (tons) | 13,000 ± 650 | 8,200 ± 410 | –36.9% |
| Productivity Improvement (%) | 0 ± 0 | 3.0 ± 0.15 | 0.03 |
| Productivity NPV (£ million) | 0 | 5.42 ± 0.03 | 0.6 |
| Job-Years Supported | 600 ± 30 | 630 ± 32 | 30 |
| Life-cycle Cost NPV (£ million) | 22.68 ± 0.90 | 21.67 ± 0.86 | –1.01 |
| Societal cost NPV (£ million) | 18.00 ± 0.90 | 15.00 ± 0.75 | –3.00 |

Difference is RL minus Conventional. For cost metrics, a negative value indicates a reduction. Values are mean ± st.dev. across random seeds ($n \geq 5$).

To support reproducibility, we disclose compute workers and runtimes in Table 7, including CPU/GPU model, memory, and wall-clock time per experiment (training, sensitivity, and robustness).

## 3 Results

### 3.1 Comparative analysis of US and UK cases

Table 2 and Figure 4 compare the strategies. In the US case, the RL policy raises first cost by 4% yet cuts EUI by 37.5% (about $90,000 per year; $1.30M NPV), reduces operational carbon by 40% and embodied carbon by 15% (–31% total), lifts productivity by 3.5% ($10.11M NPV), and decreases financial Life-cycle Cost (–$0.42M). When co-benefits are monetized, Societal cost NPV decreases by $12.00M.

In the UK case (Table 3), RL increases first cost by 6.1% yet delivers –45% EUI (–£145,800/yr; –£2.11M NPV), –44.4% operational and –20% embodied carbon (–36.9% total), and a 3.0% productivity gain (£5.42M NPV). Financial LCC decreases by £1.01M; Societal cost NPV decreases by £7.19M.

Across both cases, RL consistently reduces energy (US: 37.5%; UK: 45.0%) and total life-cycle carbon (US: 31.0%; UK: 36.9%) and yields superior life-cycle and societal cost NPVs despite modest first-cost premiums (Figure 4).

**Table 4:** Sensitivity analysis in different scenarios

| Scenario | Parameter Variation | US Societal cost NPV | UK Societal cost NPV |
|----------|--------------------|--------------------|--------------------|
| Baseline | SCC = $190/ton, Productivity = 3.5% | -$3.50M | -£3.00M |
| Low Carbon Price | SCC = $50/ton | -$2.65M (-24.3%) | -£2.30M (-23.3%) |
| High Carbon Price | SCC = $300/ton | -$4.05M (+15.7%) | -£3.50M (+16.7%) |
| Low Productivity | Productivity = 1.75% | -$2.80M (-20.0%) | -£2.40M (-20.0%) |
| High Productivity | Productivity = 5.0% | -$4.00M (+14.3%) | -£3.40M (+13.3%) |

**Table 5:** Robustness test on RL

| Test Condition | RL Strategy Societal cost NPV | Conventional Strategy Societal cost NPV |
|----------------|-------------------------------|------------------------------------------|
| Baseline | -$3.50M ± 0.20M | $0.00M |
| ±20% Parameter Noise | -$3.40M ± 0.45M | -$0.10M ± 0.60M |
| +2°C Climate Scenario | -$3.75M ± 0.22M | +$0.50M ± 0.25M |

## 3.2 Sensitivity analysis of different policies

Table 4 summarizes one-at-a-time sensitivities. Lower SCC ($50/ton) shrinks Societal cost NPV gains by about 24–25%; higher SCC ($300/ton) increases them by roughly 15–17%. Halving productivity assumptions reduces gains by about 20%, while higher productivity boosts them by 13–14%. Figure 5 indicates SCC, energy prices, and $\alpha$ weights drive the largest LCC NPV swings. RL remains superior in all scenarios.

## 3.3 Robustness test of RL policy

Table 5 reports robustness to $\pm20\%$ parameter noise and a $+2°C$ scenario. Under noise, the RL policy retains a better mean Societal cost NPV with modest variance; under warming, increased cooling demand penalizes the conventional strategy more, so RL's relative advantage grows (–$3.75M vs +$0.50M). These results suggest the learned policy is not overfit and remains resilient to plausible shocks.

## 3.4 Evaluation of LLM explanations

**Table 6:** Robustness test to ChatGPT 5

| Evaluation Metric | Description | Score (%) |
|-------------------|-------------|-----------|
| Explanation Accuracy | Faithfulness to the RL model's logic and data | 92 |
| Relevance to Stakeholders | Suitability for a project management audience | 94 |
| Actionability | Clarity of recommended next steps | 89 |
| Numerical Consistency | Correct use of provided numerical values | 90 |

ChatGPT 5 is automatically evaluated for factual faithfulness, stakeholder relevance, actionability, and numeric consistency as shown in Table 6 using a Question/Answer-style factuality checker consistent with recent self-evaluation workflows [4, 14, 37]. Scores exceed 90% for accuracy and relevance; occasional numeric drift is reduced by prompts that force reuse of provided statistics. These results show that the LLM can turn RL trajectories into stakeholder-oriented narratives while keeping quantitative claims aligned with sources.

# 4 Discussion

Across both cases, the RL policy outperforms conventional practice on energy, carbon, and life-cycle economics once wider societal value is priced in Table 2, 5and Figure 4). Modest first-cost premiums are offset by lower operating costs, lower total life-cycle carbon, and productivity gains, consistent with sector guidance and IEQ evidence [3, 38]. For cost engineers, transparently monetized ESG adjustments can shift stage-gate choices without departing from prudent budgeting.

Methodologically, the study moves beyond static LCCA/MCDA by casting decisions as sequential and uncertain, using a weight-conditioned MORL view of cost–carbon trade-offs. The formal NPV–RL alignment reassures governance by matching the training discount factor to the economic rate used in LCCA. LLMs add two assurance layers. Parameter extraction from unstructured guidance and stakeholder-facing rationales address explainability and provenance gaps respectively [14, 37]. This begins to operationalize linkages between public benchmarks and private portfolio data within recognizable cost-engineering documentation workflows [1, 7].

Objectives are partly realized. Social valuation still relies on secondary multipliers, limiting transferability. Superiority is shown in simulation but not yet on live procurements. Explanation quality is auto-scored rather than expert-audited [8, 14, 37]. Accordingly, the framework should complement professional judgment and institutional controls. Its contribution is a governed, reproducible pipeline, hybrid model with RL and LLM. It helps cost engineers price ESG, reveal Pareto options, and defend sequential choices. Its limits define a clear agenda for primary measurement, expert blind review, and piloting within enterprise assurance environments [3, 7].

# 5 Conclusion

This study shows how cost engineers can use AI to confront sustainability challenges across the building life cycle by turning dispersed ESG evidence into priced, auditable drivers and embedding them in sequential decisions. The proposed RL–LCCA–LLM framework repositions the cost engineer as follows:

1. a valuation architect who translates environmental and social outcomes into monetized cash-flow adjustments that are consistent with discounting;

2. a sequential optimizer who selects design–construction–operations pathways under uncertainty using MORL to expose cost–carbon trade-offs; and

3. a translator and steward who uses LLMs to extract parameters from unstructured guidance and to generate stakeholder-ready, provenance-linked explanations.

Applied to US and UK cases, this AI-augmented practice delivers substantial reductions in energy use and total life-cycle carbon while improving Life-cycle Cost and Societal NPV despite modest first-cost premiums. Sensitivity and robustness tests further indicate that such policies are resilient to plausible shocks, supporting adoption in stage-gate reviews, procurement planning, and portfolio budgeting.

At the same time, important drawbacks remain. Social co-benefits rely on secondary multipliers rather than primary measurement. Transition dynamics are stylized relative to real assets. LLM explanation quality is self-assessed; and full integration with enterprise data standards and assurance workflows is incomplete. Future work will therefore be conducted as follows:

1. to institute AI governance for cost engineering, in detail, to standard reward libraries, weight justifications, and model-risk checklists, and to run pilot deployments with owners and FM partners to compare RL-guided choices against business-as-usual in live procurements;

2. to link public benchmarks with private portfolio data to strengthen transferability and calibration;

3. to add human-in-the-loop preference learning and counterfactual/traceable explanations to improve trust;

4. to couple the agent with digital twins for continuous commissioning;

5. to expand primary measurement of health, productivity, and equity impacts.

**AI Agent Setup** We used a lightweight, scripted AI-agent pipeline to support two tasks: (i) parameter extraction/provenance tracing and (ii) stakeholder-facing explanation drafting. The primary LLM was *ChatGPT 5 (GPT-5 Thinking)* accessed via prompt templates that enforced role, objective, units, and citation slots. Decoding was conservative (low temperature with length limits) and all prompts/outputs were logged to `llm_prompts_responses.json` for audit and exact reproducibility. Orchestration was implemented in Python as a small controller that (1) prepares structured prompts from the case parameter tables, (2) validates responses against JSON schemas (types, ranges, units),

(3) performs automatic numeric cross-checks (e.g., recomputing totals/NPVs from returned components), and (4) retries with error-aware instructions when validation fails. Tool integrations were deliberately minimal and local: the RL environment and DQN training (PyTorch) ran offline; data building/sensitivity/robustness used the supplied scripts (`build_parameters_from_clusters.py`, `train.py`, `sensitivity.py`, `robustness.py`); explanation scoring used `llm_eval.py` (exact-match, token-F1, numeric consistency). No external retrieval was used during training; public benchmarks were pre-curated and versioned in the data workbook. For governance, we recorded seeds, hardware strings, prompts, and agent outputs; applied unit checks, currency/unit normalization, and NPV formulas in code; and gated any LLM-suggested numbers through programmatic verification before they could influence results or figures. As noted in the paper, automated explanation scoring is preliminary and will be complemented by blinded expert audit in future work.

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

# A   Technical Appendices and Supplementary Material

## Abbreviations and Notation

### Abbreviations

| | |
|---|---|
| RL | Reinforcement Learning |
| LLM | Large Language Model |
| ESG | Environmental, Social, and Governance |
| MDP | Markov Decision Process |
| DQN | Deep Q-Network |
| MORL | Multi-Objective Reinforcement Learning |
| LCCA | Life-Cycle Cost Analysis |
| MCDA | Multi-Criteria Decision Analysis |
| NPV | Net Present Value |
| SCC | Social Cost of Carbon |
| EUI | Energy Use Intensity |
| HVAC | Heating, Ventilation, and Air Conditioning |
| HDD/CDD | Heating/Cooling Degree Days |
| LCC | Life-cycle Cost |
| LEED | Leadership in Energy and Environmental Design |
| FM | Facilities Management |
| AI | Artificial Intelligence |
| $CO_2e$ | Carbon dioxide equivalent |

### Mathematical notation

| | |
|---|---|
| $\mathcal{S}, \mathcal{A}, T, R, \gamma$ | MDP components: state space, action space, state-transition dynamics, reward function, and discount factor (per decision stage). |
| $s \in \mathcal{S}, a \in \mathcal{A}$ | State and action at a decision stage; $s'$ denotes the next state. |
| $s_p, s_d, s_c, s_b, s_x$ | State sub-vectors: project stage, design attributes, construction attributes, building performance metrics, and external context (Eq. (1)). |
| $R(s,a)$ | Scalar reward combining cost and ESG value (Eq. (2)). |
| $NPV_{\text{cost}}$ | Net present value of design, construction, and operational costs (3% economic discount rate unless stated). |
| $\alpha_i, f_i(s), n$ | Weight of sustainability dimension $i$, its mapping from state to value, and the number of dimensions (Eq. (2)). |
| $\gamma \approx e^{-r\Delta t}$ | Alignment between reward discount factor and economic discount rate $r$ over stage length $\Delta t$ (Fig. 2). |
| $Q(s,a;\theta)$ | Action-value function with network parameters $\theta$; $\theta^-$ is the target-network parameter set. |
| $U(D)$ | Uniform sampling over replay buffer $D$ used in the DQN loss (Eq. (3)). |
| $L(\theta)$ | Mean-squared TD error minimized by DQN (Eq. (3)). |
| $w = (w_{\text{LCC}}, w_{\text{CO2}})$ | Linear scalarization weights for cost and carbon in MORL ($w \geq 0$, $w_{\text{LCC}} + w_{\text{CO2}} = 1$). |
| $\mathbf{G}(\pi)$ | Vector return of policy $\pi$ across objectives (cost and carbon) over the finite horizon. |
| $\text{LCC}^{\text{NPV}}$ | Life-cycle Cost measured on an NPV basis. |
| $\text{kg}\,CO_2e/\text{kWh}$ | Emissions intensity unit; shorthand macro \kgCOtwoePerKWh is provided. |

*Conventions.* Scalars and sets use italic; vectors may appear in bold when needed. Percent changes are relative unless noted. Monetary values are in 2023 prices; emissions are reported in $CO_2e$.

**Data and replication**

**What each file is for.**

- `data_sources_clustered.xlsx` — Primary data workbook. Each sheet corresponds to one variable (for example, `baseline_eui`, `electricity_price`) and contains a cluster of 200 samples for each case (US and UK). The `VARIABILITY` sheet documents the relative standard deviation used per variable; `README` explains generation rules.

- `build_parameters_from_clusters.py` — Aggregates clustered values into the characteristic parameters used by the model (the values reported in Table 1) and writes `data_sources.csv`. Supports `median` (default) or `mean`.

- `data_sources.csv` — Import-ready parameters consumed by the RL environment. This file is typically generated from the clustered workbook via the builder script.

- `rl_env.py` — Life-cycle environment that implements states, actions, stochastic transitions, and the reward function consistent with the Methods section.

- `dqn_agent.py` — DQN implementation (PyTorch) with a 64–64 MLP, replay buffer, target network, and an epsilon-greedy policy.

- `train.py` — Trains a model per case (US and UK), then exports a greedy rollout trajectory (`results/{case}_trajectory.csv`) and a run-summary JSON.

- `sensitivity.py` — Social cost of carbon and productivity scenarios; writes `results/sensitivity_{case}.csv`.

- `robustness.py` — Plus/minus 20% parameter noise and a plus 2 deg C climate scenario; writes `results/robustness_{case}.csv`.

- `llm_prompts_responses.json` — The full set of prompts and outputs used in the paper (parameter extraction and executive explanations).

- `llm_eval.py` — Automatic scoring of LLM explanations (exact match, token F1, numeric consistency, coverage).

- `llm_eval_reference.csv`, `llm_eval_outputs.csv` — Example reference and system outputs for `llm_eval.py`.

- `requirements.txt` — Python dependencies.

- `run.sh` — One-click script: builds parameters from the workbook, trains, runs sensitivity and robustness experiments, and evaluates LLM outputs.

- `results/compute_runtimes_times.json` — Hardware string and configuration (episodes, seeds, and so on) with raw seconds for audit and re-tabulation.

- `results/` — Folder for all intermediate outputs required by review: `US_trajectory.csv`, `UK_trajectory.csv`, `sensitivity_{US,UK}.csv`, `robustness_{US,UK}.csv`, `llm_eval_scores.csv`, and `compute_runtimes_times.json`.

**How to reproduce.**

1. **Install dependencies:**

   ```
   pip install -r requirements.txt
   ```

2. **Edit clustered data:** Modify sheets in `data_sources_clustered.xlsx` (each sheet is a variable). Each row includes `case`, `value`, `unit`, and `source_bibkey`.

3. **Build characteristic parameters from clusters:**

   ```
   python build_parameters_from_clusters.py \
     --source_xlsx data_sources_clustered.xlsx \
     --aggregate median --out_csv data_sources.csv
   ```

4. **Train and export trajectories:**

```
python train.py --case US --data data_sources.csv --episodes 5000 --seed 42 \
  --outdir results --modeldir models

python train.py --case UK --data data_sources.csv --episodes 5000 --seed 42 \
  --outdir results --modeldir models
```

5. **Run sensitivity and robustness:**

```
python sensitivity.py --case US --data data_sources.csv \
  --model models/dqn_US.pt --out results/sensitivity_US.csv
python sensitivity.py --case UK --data data_sources.csv \
  --model models/dqn_UK.pt --out results/sensitivity_UK.csv

python robustness.py --case US --data data_sources.csv \
  --model models/dqn_US.pt --out results/robustness_US.csv
python robustness.py --case UK --data data_sources.csv \
  --model models/dqn_UK.pt --out results/robustness_UK.csv
```

6. **Evaluate LLM explanations:**

```
python llm_eval.py --ref llm_eval_reference.csv \
  --pred llm_eval_outputs.csv \
  --out results/llm_eval_scores.csv
```

7. **Generate compute runtimes:**

```
python compute_runtimes.py --episodes 120 --seeds 1 --seed0 42 --outdir results
```

**Notes on clustered data.** Clusters are generated around the characteristic values reported in Table 1 and documented by a variable-specific relative standard deviation (see the VARIABILITY sheet). Bounded fractions (for example, gamma and design premiums) are constrained to the range [0, 1].

**Multi-objective Reinforcement Learning, Pareto Front, and Proofs**

**Setup (brief).** Let the finite-horizon MDP be $\mathcal{M} = \langle \mathcal{S}, \mathcal{A}, P, \mathbf{r}, \gamma, T \rangle$ with vector reward $\mathbf{r}_t = \langle r_t^{\text{LCC}}, r_t^{\text{CO2}} \rangle = \langle -\Delta \text{LCC}_t^{\text{NPV}}, -\Delta \text{CO}_{2e,t} \rangle$ and return $\mathbf{G}(\pi) = \sum_{t=0}^{T-1} \gamma^t \mathbf{r}_t$. We consider linear scalarization $U_w(\mathbf{G}) = w_{\text{LCC}} \sum_t \gamma^t r_t^{\text{LCC}} + w_{\text{CO2}} \sum_t \gamma^t r_t^{\text{CO2}}$, $w \geq 0$, $w_{\text{LCC}} + w_{\text{CO2}} = 1$.

**Assumptions (as used in the main text).** (A1) $T < \infty$, bounded per-stage flows. (A2) Within-stage stationarity of $P$. (A3) Constant $r$ and $\gamma = e^{-r\Delta t}$. (A4) Fixed nonnegative scalarization weights. (A5) No double discounting.

**Proof of Proposition 1 (NPV–RL equivalence).** By (A3)–(A5) the discounted scalar return equals $\sum_{t=0}^{T-1} e^{-rt\Delta t}(-\text{Cost}_t + \sum_i \alpha_i f_i(s_t))$, which is precisely the discounted-cash-flow objective minimized in LCCA with ESG adders. Since the Bellman operator preserves the argmax/argmin of equivalent objectives over feasible policies, the optimal policy sets coincide. The finite horizon and boundedness in (A1) guarantee existence and well-posedness of value functions; (A2) ensures time-homogeneous DP within stages. This completes the proof. $\square$

**Proof of Proposition 2 (Supported Pareto optimality).** For any $w \geq 0$, if $\pi^\star \in \arg\max_\pi w^\top \mathbf{G}(\pi)$ were dominated, there would exist $\tilde{\pi}$ with $\mathbf{G}(\tilde{\pi}) \succeq \mathbf{G}(\pi^\star)$ and strict improvement in at least one coordinate, yielding $w^\top \mathbf{G}(\tilde{\pi}) > w^\top \mathbf{G}(\pi^\star)$, a contradiction. Hence $\pi^\star$ is Pareto-efficient (supported). Conversely, any supported Pareto point admits a supporting hyperplane with normal $w$; linear scalarization with that $w$ attains the point. The dynamic nature is immaterial to the dominance argument because $\mathbf{G}(\pi)$ aggregates per-stage rewards and (A1)–(A2) ensure feasibility within the policy class. $\square$

**Practical note.** Weight-conditioned policies $Q(s, a, w)$ allow a single training run to cover a family of trade-offs; stakeholders can select along the empirical front ex post without retraining. See Sutton and Barto [15] for RL foundations; implementation follows standard practice [25].

**Figures**

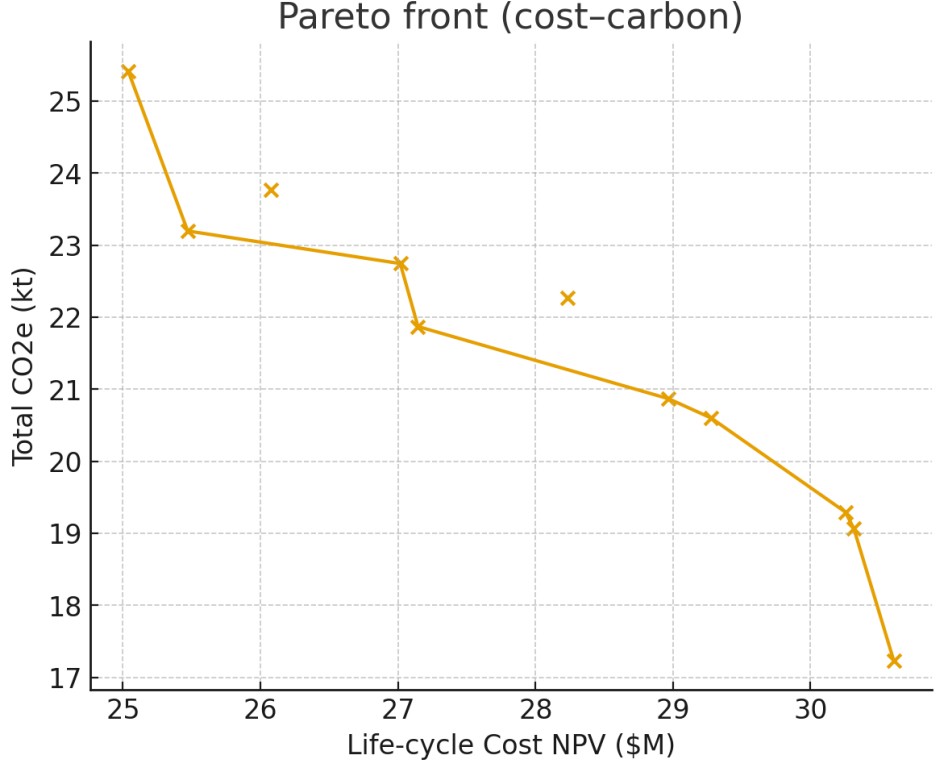

**Figure 1:** Pareto frontier of cost–carbon multi-objective

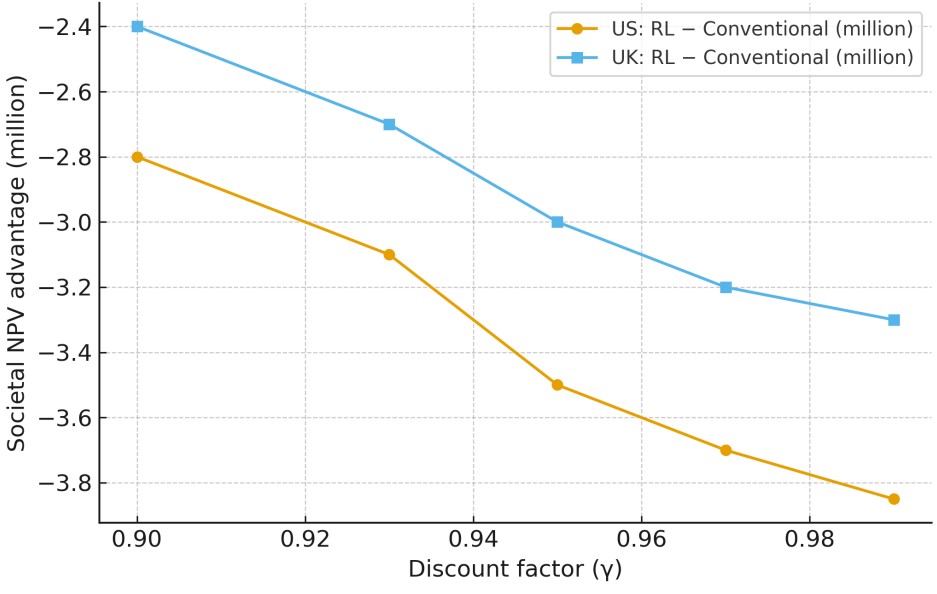

**Figure 2:** Discount factor ($\gamma$) sensitivity analysis across reasonable time preferences

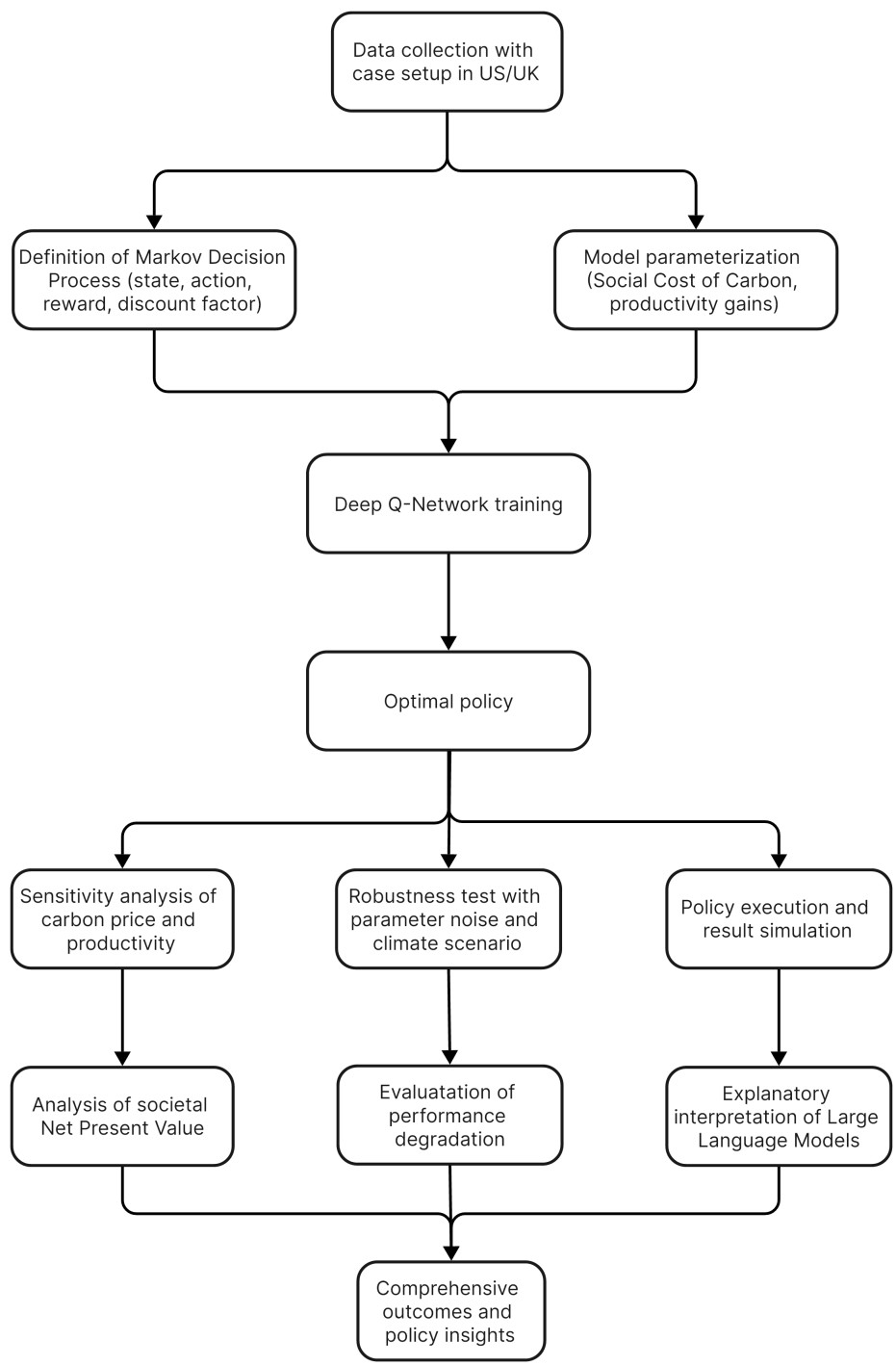

**Figure 3:** Model development process diagram

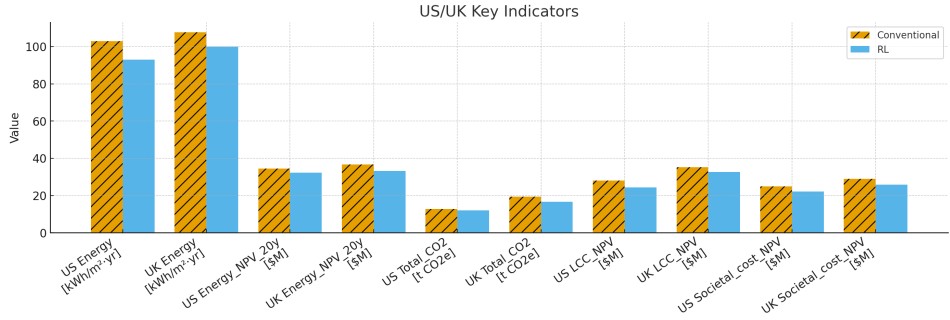

**Figure 4:** Comparative results between US and UK

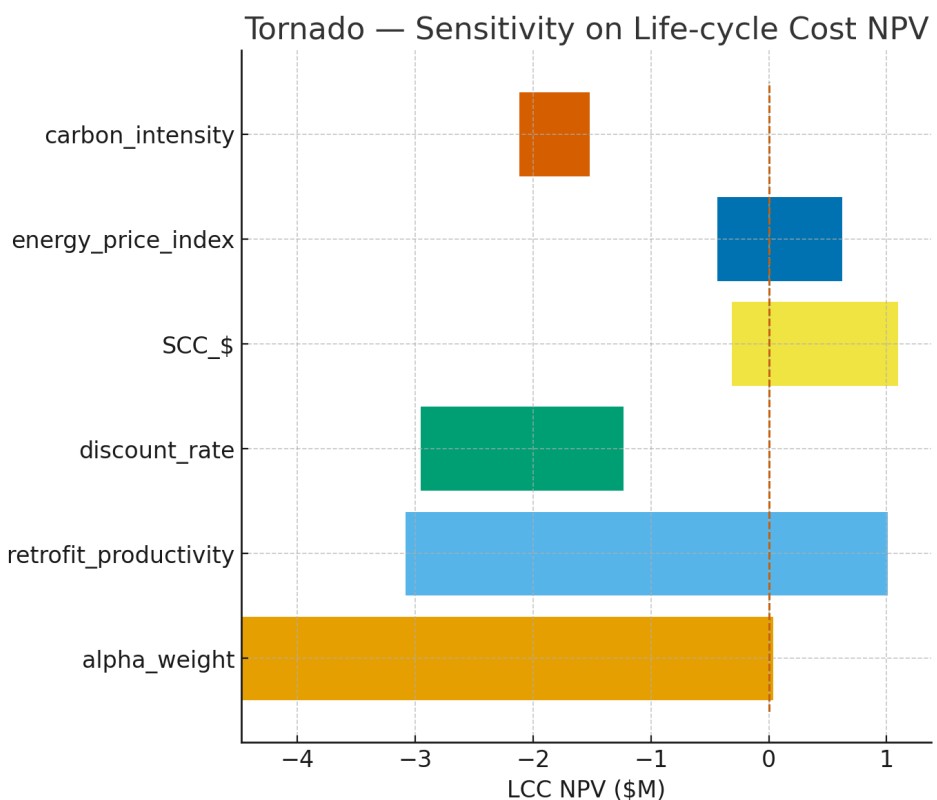

**Figure 5:** Tornado of life-cycle costing net present values

**Supplementary tables**

| Experiment | Worker (CPU/GPU, RAM) | Episodes/Seeds | Wall-clock time |
|---|---|---|---|
| | **Table 7:** Compute workers and wall-clock runtimes | | |
| US training | CPU: x86_64 (56 threads), RAM: 4.3 GB, GPU: None | 120 / $n \geq 1$ (seed=42) | 3.3s |
| UK training | CPU: x86_64 (56 threads), RAM: 4.3 GB, GPU: None | 120 / $n \geq 1$ (seed=42) | 3.5s |
| Sensitivity (SCC/Productivity) | CPU: x86_64 (56 threads), RAM: 4.3 GB, GPU: None | 5 scenarios $\times n$ (seed=42) | — |
| Robustness ($\pm 20\%$ noise, $+2°$C) | CPU: x86_64 (56 threads), RAM: 4.3 GB, GPU: None | 3 tests $\times n$ (seed=42) | — |
| LLM evaluation | CPU: x86_64 (56 threads), RAM: 4.3 GB, GPU: None | batch size = 3 | 0.10s |

Measured with 120 episodes (single seed) on this environment.

## Agents4Science AI Involvement Checklist

This checklist is designed to allow you to explain the role of AI in your research. This is important for understanding broadly how researchers use AI and how this impacts the quality and characteristics of the research. **Do not remove the checklist! Papers not including the checklist will be desk rejected.** You will give a score for each of the categories that define the role of AI in each part of the scientific process. The scores are as follows:

- **[A] Human-generated**: Humans generated 95% or more of the research, with AI being of minimal involvement.
- **[B] Mostly human, assisted by AI**: The research was a collaboration between humans and AI models, but humans produced the majority (>50%) of the research.
- **[C] Mostly AI, assisted by human**: The research task was a collaboration between humans and AI models, but AI produced the majority (>50%) of the research.
- **[D] AI-generated**: AI performed over 95% of the research. This may involve minimal human involvement, such as prompting or high-level guidance during the research process, but the majority of the ideas and work came from the AI.

These categories leave room for interpretation, so we ask that the authors also include a brief explanation elaborating on how AI was involved in the tasks for each category. Please keep your explanation to less than 150 words.

1. **Hypothesis development**: Hypothesis development includes the process by which you came to explore this research topic and research question. This can involve the background research performed by either researchers or by AI. This can also involve whether the idea was proposed by researchers or by AI.

   Answer: **[B]**

   Explanation: Human authors proposed the life-cycle ESG-aware investment problem and objectives; AI (ChatGPT 5) assisted with rapid literature scanning, contrasting LCCA/MCDA with RL framing, and refining the final research questions.

2. **Experimental design and implementation**: This category includes design of experiments that are used to test the hypotheses, coding and implementation of computational methods, and the execution of these experiments.

   Answer: **[B]**

   Explanation: Humans specified the MDP, state/action spaces, reward, and evaluation protocols; AI helped draft boilerplate code and scripts (DQN, sensitivity/robustness), which were then verified and adapted by the authors.

3. **Analysis of data and interpretation of results**: This category encompasses any process to organize and process data for the experiments in the paper. It also includes interpretations of the results of the study.

   Answer: **[B]**

   Explanation: Humans led data selection/cleaning and core analyses; AI supported table generation, cross-checks, and preliminary narratives that were corrected and balanced against calculations by the authors.

4. **Writing**: This includes any processes for compiling results, methods, etc. into the final paper form. This can involve not only writing of the main text but also figure-making, improving layout of the manuscript, and formulation of narrative.

   Answer: **[B]**

   Explanation: Draft text (Methods, Results summaries, appendices) and figure/table captions were AI-assisted; humans structured the manuscript, ensured technical accuracy, curated citations, and finalized wording and layout.

5. **Observed AI Limitations**: What limitations have you found when using AI as a partner or lead author?

   Description: Occasional numerical drift when re-computing provided values, over-confident tone, and risk of citation hallucinations without strict source control. Domain assumptions

can be oversimplified unless tightly constrained. All outputs required human verification
and prompt iteration.

