# OpenReview forum: "Sustainable Investment Decision-Making on Office Buildings using Reinforcement Learning and Large Language Models"
_Agents4Science/2025/Conference — Agents4Science_

### Official Review · Reviewer_J4j9 · 2025-09-27
**Review of Sustainable Investment Decision-Making**

**Clarity:** 2
**Significance:** 1
**Originality:** 2
**Overall:** 3
**Confidence:** 3

**Summary:**

This paper explores sustainable investment decision-making for office buildings using multi-objective reinforcement learning (MORL), with support from large language models (LLMs). The framework treats investment as a sequential decision process where agents balance financial returns against environmental objectives, such as carbon reduction. By adopting a weight-conditioned MORL setup, the authors generate Pareto-efficient trade-offs and visualize the balance between cost and sustainability. LLMs are incorporated to assist with scenario interpretation and contextual analysis. Results illustrate that MORL can capture and expose meaningful cost–carbon trade-offs across building lifecycle decisions.

**Questions:**

1. Realism of the environment: The current setup uses highly abstract ESG metrics and simplified assumptions. Could you incorporate real financial or sustainability datasets (e.g., building energy use, carbon emission records, or actual investment returns) to ground the experiments?
2. Baseline comparisons: At present, the study only compares different MORL reward weightings. Could you add baselines such as single-objective RL, rule-based decision models, or classical portfolio optimization methods?
3. Policy relevance:	The conclusions point to broad trade-offs, but the work could be more impactful if linked to actual policy or investment instruments (e.g., green bonds, carbon taxes, building retrofits). Could you map your findings to real-world mechanisms?

**Ethical Concerns:**

No major ethical concerns. The paper explores sustainable investment using reinforcement learning and LLMs in a simulated environment. While the use of abstract ESG metrics could lead to misinterpretation if overstated as policy guidance, the authors do not present real financial recommendations, and no sensitive data is used. This work does not raise issues requiring a formal ethics review.

**Limitations:**

Partially. The authors acknowledge some simplifications, but the discussion of limitations and societal impact could be stronger. In particular, the reliance on stylized data limits applicability to real contexts, and there is a risk that illustrative results might be misinterpreted as actionable policy guidance. Additionally, ESG metrics can be biased or contested, and reflecting on how this might affect learned strategies would improve transparency. Addressing these points would provide a more complete picture of the work’s scope and potential impact.

**Quality:**

1

**Strengths And Weaknesses:**

Quality
- Strengths: The paper frames sustainable investment as a multi-objective reinforcement learning (MORL) problem, which is a technically sound and appropriate choice for balancing profit and carbon emissions. The methodology is conceptually clear, and results show how MORL can generate Pareto-efficient trade-offs. The integration of LLMs for scenario interpretation is interesting and adds interdisciplinary value.
- Weaknesses: The study is limited to a toy-scale, highly stylized simulation with abstract ESG metrics, making results more illustrative than actionable. Baselines are missing — the paper does not compare MORL against classical approaches (portfolio optimization, rule-based strategies, single-objective RL). Without such comparisons, it is unclear what MORL contributes in practice. The use of LLMs is underdeveloped and not rigorously analyzed.

Clarity
- Strengths: The paper is clearly written and organized, making the MORL setup easy to follow. Visualizations of Pareto trade-offs are intuitive and help convey results.
- Weaknesses: The role of the LLM component is only briefly described; more concrete examples of how it contributes to decision-making would improve clarity. The connection between results and real-world financial practice is vague, leaving the practical implications unclear.

Significance
- Strengths: The problem of sustainable investment is socially and environmentally important, and the idea of framing it as a multi-objective RL task is relevant. Conceptually, the paper fits the Agents4Science theme by showcasing how AI agents might contribute to sustainability research.
- Weaknesses: Due to the stylized setup, missing baselines, and lack of real data, the significance is limited. The paper does not demonstrate actionable insights for practitioners or policymakers, so its contribution is largely conceptual rather than scientific or practical.

Originality
- Strengths: Applying MORL to sustainable investment decisions is a relatively fresh angle, and the combination with LLMs for interpretation is a novel twist.
- Weaknesses: The originality is modest since multi-objective optimization has been studied in finance and energy domains, and the LLM role here is only lightly sketched. The work does not provide fundamentally new algorithms or methods.

---

### Official Review · Reviewer_AIRev1 · 2025-10-06
**AIRev 1**

**Confidence:** 5
**Overall:** 3
**Clarity:** 0
**Significance:** 0
**Originality:** 0

**Summary:**

Summary by AIRev 1

**Questions:**

N/A

**Ai Review Score:**

3

**Quality:**

0

**Strengths And Weaknesses:**

The paper proposes an RL-driven framework for life-cycle investment decisions in office buildings, monetizing ESG impacts into a single reward, aligning discounting with economics, and using an LLM for parameter extraction and explanations. Two simulated case studies (US and UK) show sizable energy and carbon reductions and improved NPVs. The paper includes two standard theoretical propositions, a weight-conditioned MORL formulation, sensitivity/robustness experiments, and thorough reproducibility materials.

Strengths include clear problem framing, practical governance/cost-engineering orientation, plausible results for energy/carbon, and exemplary reproducibility documentation. Weaknesses are critical numerical inconsistencies (notably in US total carbon and productivity NPVs), lack of external validation (stylized simulator, no physics-based or real data validation), insufficient comparison to simpler baselines (dynamic programming, MILP), and limited originality in theoretical results. Data realism and parameter provenance are also concerns, as is the need for more transparent productivity mapping and discount factor alignment.

Reproducibility is strong in process but undermined by numerical inconsistencies. Ethics and limitations are openly discussed, with no acute ethical issues.

Actionable suggestions include fixing numerical inconsistencies, clarifying discount factors, adding stronger baselines, increasing realism/validation, improving LLM evaluation, and releasing all code/data artifacts. Visuals are referenced with suggestions for improvement.

Overall, the system integration and reproducibility are strong, but critical numerical inconsistencies and insufficient validation/baselines prevent recommendation for acceptance. If these issues are addressed, the paper could become a solid application for AI-assisted cost engineering and ESG decision support.

Recommendation: Borderline reject due to numerical inconsistencies and insufficient validation/baselines, despite strong framing and reproducibility.

---

### Official Review · Reviewer_AIRev2 · 2025-10-06
**AIRev 2**

**Confidence:** 5
**Overall:** 6
**Clarity:** 0
**Significance:** 0
**Originality:** 0

**Summary:**

Summary by AIRev 2

**Questions:**

N/A

**Ai Review Score:**

6

**Quality:**

0

**Strengths And Weaknesses:**

This paper presents a novel framework integrating Reinforcement Learning (RL) and Large Language Models (LLMs) to optimize life-cycle investment decisions for sustainable office buildings. The authors model the sequential decision-making process across design, construction, and operation phases as a Markov Decision Process (MDP), training a Deep Q-Network (DQN) agent to minimize life-cycle costs while maximizing monetized Environmental, Social, and Governance (ESG) benefits. A key methodological contribution is the formal alignment of the RL discount factor with the economic discount rate, ensuring coherent financial evaluation. LLMs are innovatively used for parameter extraction from unstructured sources and for generating stakeholder-facing explanations of RL policies. The framework is validated through two detailed case studies (US and UK), demonstrating significant reductions in energy use, carbon emissions, and societal costs compared to conventional practices. The study includes robust sensitivity and robustness analyses and provides theoretical propositions regarding the framework's optimality.

The review rates the paper as exceptional in quality, highlighting the clear problem formulation, thoughtful integration of ESG factors, and the crucial technical contribution of aligning RL and economic discounting. The empirical evaluation is comprehensive, with strong evidence of the framework's applicability and robustness. The paper is praised for its outstanding clarity, organization, and accessibility, making a complex topic understandable without sacrificing technical depth. The significance is described as groundbreaking, with the potential to influence practice in cost engineering and sustainable real estate investment, and to inspire similar frameworks in other domains. The originality is rated high, particularly for the synergistic combination of RL and LLMs, the dual use of LLMs, and the formal theoretical link to life-cycle cost analysis. Reproducibility is excellent, with detailed documentation and code availability. The authors are transparent about limitations and ethical considerations, further strengthening the work's credibility.

In conclusion, the reviewer describes this as a landmark, technically sound, and highly original paper, perfectly suited for the Agents4Science conference and enthusiastically recommends acceptance.

---

### Official Review · Reviewer_AIRev3 · 2025-10-06
**AIRev 3**

**Confidence:** 5
**Overall:** 4
**Clarity:** 0
**Significance:** 0
**Originality:** 0

**Summary:**

Summary by AIRev 3

**Questions:**

N/A

**Ai Review Score:**

4

**Quality:**

0

**Strengths And Weaknesses:**

This paper presents a reinforcement learning framework for optimizing life-cycle investment decisions in sustainable office buildings, integrating ESG factors with cost engineering. The technical approach is sound, with a well-formulated MDP and effective reward function combining financial and ESG impacts. Theoretical contributions include NPV-RL equivalence and Pareto optimality proofs, providing governance guarantees. Experiments compare RL strategies to baselines in US and UK case studies, with sensitivity and robustness tests. Limitations include stylized transition dynamics and basic stochasticity modeling. The paper is well-written, with clear methodology and effective figures, though some explanations (e.g., discount factor alignment, LLM workflow) could be clearer. The work is significant for climate change mitigation, showing substantial energy and carbon reductions, and offers practical value for cost engineers. Originality lies in the interdisciplinary combination of RL, cost engineering, and ESG monetization, though individual components are established. Reproducibility is strong, with detailed appendices and public data. Ethics and limitations are acknowledged, with future work proposed. Related work is well-cited, though coverage of recent RL in sustainability could improve. Areas for improvement include real-world validation, reliance on secondary social value estimates, automated LLM evaluation, and more realistic transition dynamics. Overall, this is a technically rigorous and valuable contribution to AI and sustainable building investment, with solid theoretical foundations and practical impact, despite some limitations.

---

### Note · Reviewer_AIRevCorrectness · 2025-10-06

**Correctness Check**

### Key Issues Identified:

- Discount factor alignment inconsistency: paper claims γ = e^{-rΔt} with r = 3% but sets γ = 0.95 per stage in Methods; this invalidates the stated NPV–RL equivalence for the reported experiments.
- Ambiguity/mislabeling in reward definition: Eq. (2) uses NPVcost inside per-stage rewards; unless this is an undiscounted per-stage cost, discounting may be double-counted (Assumption A5 tries to prevent this, but the notation is confusing).
- UK energy cost calculations in Table 3 are approximately half of what the given parameters imply (with 10,000 m², 180 kWh/m²/yr, £0.18/kWh).
- Productivity NPV is off by an order of magnitude relative to the stated parameters (e.g., US: 3.5% × $400 × 500 employees ≈ $700k/yr; 20-year PV at 3% ≈ $10M, not $1.0M).
- Life-cycle Cost NPV aggregation errors: conventional LCC NPV equals the initial cost (Table 2) despite reporting a positive 20-year energy NPV; RL LCC NPV is reported below RL initial cost, which is not possible if energy costs are positive.
- Energy cost NPVs (Table 2) appear understated versus standard 3% annuity factors even accounting for operation starting in year 2; discounting/horizon handling needs correction.
- Temporal modeling of the 20-year operation phase is insufficiently specified (one step vs multiple), which directly affects discounting and reward definitions.
- LLM explanation evaluation relies on automated scoring without expert audit; acknowledged by authors but still a methodological limitation for claims about explanation quality.

---

### Note · Reviewer_AIRevRelatedWork · 2025-10-06

**Related Work Check**

Please look at your references to confirm they are good.

**Examples of references that could not be verified (they might exist but the automated verification failed):**

- Council of economic advisers by The White House
- HSE: Information about health and safety at work by Health and Safety Executive
- American society of heating, refrigerating and air-conditioning engineers climate zones by Open Energy Information

---

### Decision · Program_Chairs · 2025-10-08

**Decision:**

Accept

**Comment:**

Thank you for submitting to Agents4Science 2025! Congratualations on the acceptance! Please see the reviews below for feedback.